# Virtual Network Function Migration Considering Load Balance and SFC Delay in 6G Mobile Edge Computing Networks

Yi Yue [1,*] , Xiongyan Tang [1], Zhiyan Zhang [1], Xuebei Zhang [1] and Wencong Yang [1,2]

1   Future Network Research Department, China Unicom Research Institute, Beijing 100048, China; tangxy@chinaunicom.cn (X.T.); zhangxb170@chinaunicom.cn (X.Z.); yangwc27@chinaunicom.cn (W.Y.)
2   Electrical Engineering Department, Zhengzhou University, Zhengzhou 450001, China
*   Correspondence: yuey80@chinaunicom.cn

**Abstract:** With the emergence of Network Function Virtualization (NFV) and Software-Defined Networks (SDN), Service Function Chaining (SFC) has evolved into a popular paradigm for carrying and fulfilling network services. However, the implementation of Mobile Edge Computing (MEC) in sixth-generation (6G) mobile networks requires efficient resource allocation mechanisms to migrate virtual network functions (VNFs). Deep learning is a promising approach to address this problem. Currently, research on VNF migration mainly focuses on how to migrate a single VNF while ignoring the VNF sharing and concurrent migration. Moreover, most existing VNF migration algorithms are complex, unscalable, and time-inefficient. This paper assumes that each placed VNF can serve multiple SFCs. We focus on selecting the best migration location for concurrently migrating VNF instances based on actual network conditions. First, we formulate the VNF migration problem as an optimization model whose goal is to minimize the end-to-end delay of all influenced SFCs while guaranteeing network load balance after migration. Next, we design a Deep Learning-based Two-Stage Algorithm (DLTSA) to solve the VNF migration problem. Finally, we combine previous experimental data to generate realistic VNF traffic patterns and evaluate the algorithm. Simulation results show that the SFC delay after migration calculated by DLTSA is close to the optimal results and much lower than the benchmarks. In addition, it effectively guarantees the load balancing of the network after migration.

**Keywords:** network function virtualization; service function chain; virtual network function migration; QoS guarantee

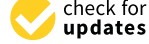



## 1. Introduction

Unlike traditional networks, future networks need to support multiple services in an easily scalable and flexible manner. Network Functions Virtualization (NFV) has emerged as a promising approach to powering future networks. NFV technology can decouple network functions from specific physical devices through virtualization technology. Network functions are implemented as Virtual Network Functions (VNFs, in software) and run on commercial-off-the-shelf devices. Therefore, cloud service providers can flexibly deploy VNFs at appropriate locations in the network and then provide customized services to users [1]. Generally, network services in NFV-enabled networks are accomplished by Service Function Chains (SFCs) [2], which consist of a sequence of specified VNFs in a predefined order [3]. We regard a user's request for SFC service as an SFC request, simplified as SFCR. Meanwhile, we call the elements in a specific SFCR as VNF requests (VNFRs), corresponding to the VNFs in an SFC.

However, opportunities and challenges always coexist. The flexible deployment of network services leads to more frequent dynamic fluctuations in the network load. When SFC provides services, traffic needs to be steered to traverse through several nodes, which may cause network nodes and links to be overloaded and violate the delay constraints

of services. According to Cisco Visual Network Index [4], mobile data traffic will attain a compound annual growth rate of 54% annually from 2016 to 2023. However, in the conventional network structure, network services are provided by dedicated hardware, which cannot effectively address the exponential increase of various service demands. Meanwhile, to cope with the development of new services, it is necessary to constantly install and maintain new dedicated equipment for Cloud Service Providers (CSPs), which rapidly increases the load and energy consumption of the data center network [5].

It should be noted that the traffic information of network services in the VNF instance is constantly updated and accumulated. Simple remapping methods cannot effectively guarantee the continuity of network service traffic and lead to massive reconfiguration expenses [6]. Therefore, compared to VNF remapping, the VNF migration mechanism is a more suitable solution. At the same time, related research has also designed a variety of algorithms to solve the migration problem of VNF in the network supporting NFV. However, some essential factors still have not been fully considered. First, VNF instance sharing is a mechanism that is common in NFV-enabled networks. Most previous studies assume that one VNF instance only provides services for one SFC and ignores the sharing mechanism. Since only the influence of migration on a single SFC needs to be considered, the VNF migration solution is relatively uncomplicated. Second, the dynamic characteristics of services in NFV networks can lead to the simultaneous overloading of multiple nodes or links, which means that VNF migrations are usually conducted concurrently. But little literature has paid attention to this issue.

Simultaneously, with significant performance improvements, deep learning has made remarkable achievements in many fields, such as natural language processing, autonomous driving, computer vision, etc., [7]. Since VNF migration is related to network conditions, it is necessary to dig deeply into the hidden rules behind the calculation of the migration scheme to improve the efficiency of selecting the optimal path during the migration process. Based on the powerful learning ability and performance optimization of software and hardware [8], deep learning can effectively discover and characterize the structural features of complex problems. Furthermore, given the scale of training data and the goal of fine-grained feature extraction and classification, deep learning is more suitable than traditional machine learning algorithms to achieve routing path calculation after VNF migration [9]. By considering many network-related factors, parameters, and metrics, deep learning can design fine-grained migration strategies for VNFs more autonomously and intelligently. It hopes to seamlessly merge intelligence and network technology by incorporating deep learning. This would minimize the need for extensive iterations and calculations when conducting intelligent routing path calculations, potentially surpassing conventional rule-based routing algorithms.

Figure 1 illustrates the integration of VNF migration and deep learning technologies to enable intelligent routing path computation for SFCRs. The controller collects network condition data on time, which is then used to train deep models. These models generate optimal strategies for efficient network management.

Numerous VNF migration models and algorithms have been studied, but some issues are yet to be resolved.

- First, in NFV-enabled environments, VNF instances are commonly shared by multiple SFCs for efficient use of network resources. However, most previous studies on VNF migration have ignored this aspect and assumed that each VNF instance is only used by a single SFC. This simplifies the design of VNF migration schemes, as only the impact on one SFC needs to be considered.
- Another issue with dynamic network services is the possibility of multiple physical links or nodes overloading simultaneously. VNF instance migration is often performed concurrently to address this problem. However, there is limited literature on this particular problem.
- Finally, most solutions for VNF migration in SFCRs operate using rule-based algorithms and cannot execute intelligent migrations. This often results in the need for

complicated strategy development and reduced efficiency when computing routing paths.

Given these facts, we consider VNF sharing and concurrent migration and introduce deep learning technology in our study of the VNF migration problem. We focus on the impact on the network and services after the migration. To ensure a more resilient network in the face of potential traffic changes, we aim to achieve network resource load balancing and minimize end-to-end delay for all affected services. The contributions of our work are listed below:

- The VNF migration problem is formulated as a mathematical model. Since VNF instance sharing is considered, one VNF may serve multiple SFCs. It means that migrating one VNF instance may affect all SFCs that use it, making our work different from most current work. While the problem is more complex, it is more in line with the requirements of real cloud datacenter networks.
- We propose a Deep-Learning-based Two-Stage Algorithm (DLTSA) to solve the problem. This algorithm comprises two components: a hybrid genetic evolution algorithm and a running algorithm. The former generates training data using available resources and various SFCRs, while the latter handles the migration of VNFs based on the gathered training data.
- We conduct a detailed analysis of DLTSA and evaluate DLTSA in cloud datacenter networks of different sizes. The performance evaluation results show that our proposed solution can effectively guarantee the network load balance after VNF migration. Additionally, it can provide a lower SFC average delay after migration than the benchmark.

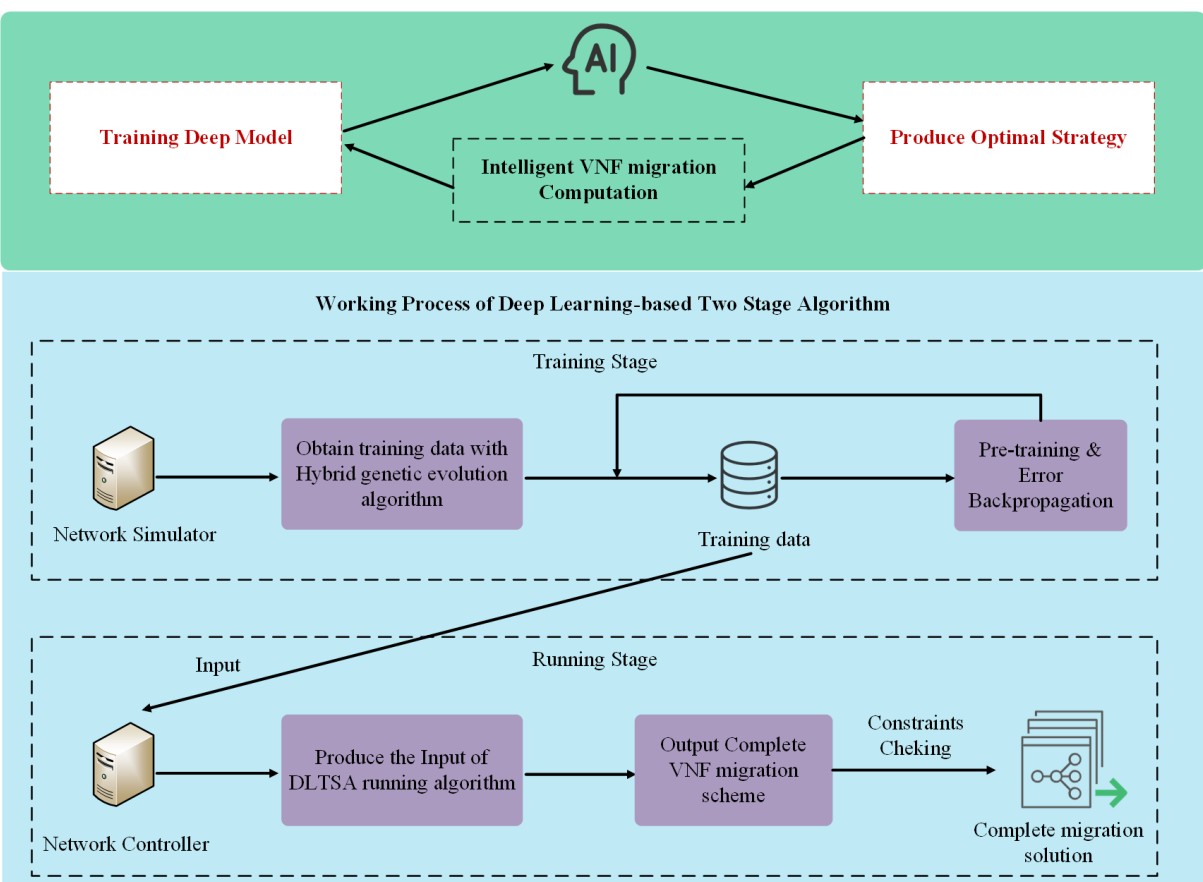

**Figure 1.** VNF migration with deep learning technology in 6G MEC networks.

We organize the remainder of this paper as follows. Section 2 reviews the related works. In Section 3, we describe the system model. Section 4 formulates the VNF migration problem and our solution DLTSA is introduced in Section 5. Afterwards, Section 6 is the performance evaluation of DLTSA. Finally, we conclude the paper in Section 7.

## 2. Related Works

This paper focuses on the VNF migration and SFC reconfiguration problem in dynamic NFV-enabled networks. There have been various algorithms proposed in the literature to solve this problem. We have reviewed the previous work and highlighted the optimization brought by our approach. To better demonstrante the contribution of this paper, a detailed comparison of related works is given in Table 1.

**Table 1.** The comparison of related works.

| Literature / Scopes | [10] | [11] | [12] | [13] | [14] | [9,15] | [16] | Our Approach |
|---|---|---|---|---|---|---|---|---|
| VNF sharing consideration | ✓ | × | × | × | × | ✓ | × | ✓ |
| Concurrent migration of multiple VNFs | × | × | ✓ | ✓ | × | × | × | ✓ |
| Applying Deep learning | × | × | × | × | × | ✓ | ✓ | ✓ |
| Network load maintenance | ✓ | ✓ | ✓ | × | ✓ | × | × | ✓ |
| SFCR delay guarantee | × | ✓ | × | ✓ | × | ✓ | × | ✓ |
| Node resource consideration | ✓ | ✓ | ✓ | ✓ | ✓ | ✓ | ✓ | ✓ |
| Link resource consideration | ✓ | ✓ | ✓ | × | × | ✓ | ✓ | ✓ |

✓ indicates that the attribute is provided in the research work, × is the opposite.

The migration and reconfiguration mechanism of NFV involves mapping VNF and resource allocation. Many studies have also focused on this aspect. For instance, the authors in [10] aimed to better utilize the network by cherry-picking the relation between VNFI sharing and link usage. They proposed a chain deployment algorithm to find a better solution. Similarly, Agarwal et al. [17] designed a queuing-based system model to realize optimal resource allocation between VNFs. Pham et al. [18] studied the VNF resource allocation problem with a sampling-based Markov approximation approach and proposed a matching algorithm based on Markov approximation to solve it in a short convergence time. The authors in [11] presented a self-adaptive VNF embedding algorithm. This adaptive algorithm divides SFC requests into different types and solves them by an integer linear programming formula. Huang et al. [16] introduced an enhanced model for mapping SFC requests. They utilized the deep deterministic policy gradient (DDPG) approach to optimize the service cost rate and mapping rate, aiming to find the best mapping strategy for the network. Additionally, they developed four VNF orchestration strategies to improve the matching accuracy for various networks based on factors such as the VNF request rate and mapping rate. In [19], the authors introduced a reconfiguration algorithm for VNF that saves energy by predicting short-term resource requirements (RP-EDM). They utilized LSTM to forecast VNF resource needs ahead of time, reducing the delay in dynamic migration and determining the optimal migration timing. In their study, Tseng et al. [14] suggested a genetic algorithm that utilizes prediction and considers the CPU and memory utilization of PM to optimize the use of network resources. The algorithm has a multi-objective approach. However, the solutions mentioned are based on rules; they need to be

capable of achieving intelligent VNF migration for SFCRs. This migration typically requires designing complex strategies and computing routing paths with low time efficiency.

In the work of [12,20], the migration problem is addressed to minimize overall energy consumption. The authors proposed a multi-heuristic approach based on the Viterbi algorithm to determine when and where to migrate VNFI. Authors in [13,21] focused on implementing migration strategies in edge networks to minimize the end-to-end delay of SFCs. Cziva et al. [13] primarily concentrated on VNF migration timing, designing a dynamic rescheduling method based on the theory of optimal stop. Meanwhile, Song et al. [21] computed the optimal number of clusters in edge networks and used a graph partitioning algorithm to minimize the number of VNF migrations between groups. While these studies have their benefits, they all assumed that one VNF is used only by one SFC, and a single VNF migration is made each time. In [22], the authors' proposed approaches query the promising embedded deep reinforcement learning engine in the management layer (e.g., orchestrator) to observe the state features of VNFs, apply the action on instantiating and modifying new/created VNFs, and evaluate the average transmission delays for end-to-end IoT services. Pei et al. [15] focused on solving the VNF selection and chaining problem in SDN/NFV-enabled networks. They developed a unique algorithm that uses deep learning to make VNF selection and chaining for SFCRs to be more intelligent and efficient. In their study, Prohim et al. [23] introduced a decentralized SDN controller that acts as an agent to facilitate a software-based system and communicate with the virtualization environment. The architecture, which includes an agent controller and orchestrator, provides a centralized view for the placement of virtual network functions. It efficiently maps virtual machines to carry out edge update procedures and create effective forwarding paths. Our study focuses on VNF sharing and concurrent migration, both significant and underexplored. We aim to minimize the end-to-end delay of all affected SFCs while guaranteeing network load balance.

## 3. System Model

### 3.1. Network Model

We represent the physical network as an undirected graph $G = (N, L)$, where $N$ and $L$ indicate the set of physical nodes and links, respectively. Specifically, we use $n_i$ to denote the $i$-th physical node and $l_j$ to indicate the $j$-th physical link. The parameters $C_{n_i}^{cpu}$ and $C_{n_i}^{mem}$ stand for the CPU capacity and memory capacity of node $n_i$. Each physical node can hold multiple VNF instances to support various VNF request (VNFRs). The bandwidth capacity of link $l_j$ is symbolized as $B_{l_j}$. We denote $D_{l_j}$ as the propagation delay on link $l_j$.

### 3.2. SFC Requests

The parameter $R = \{r_u | u \in [1, |R|]\}$ indicates the set of total SFC requests (SFCRs). A 3-tuple, $\{F^u, \lambda_u, D_u^{max}\}$ is used to indicate SFCR $r_u$. The notation $F^u = \{f_v^u | v \in [1, |I_u|]\}$ is the set of VNFRs in SFCR $r_u$, and $f_v^u$ is the $v$-th VNFR. $\lambda_u$ represents the arrival rate (in bits/sec) of SFCR $r_u$. The maximum tolerated delay of SFCR $r_u$ is symbolized as $D_u^{max}$.

We define $E^u = \{e_h^u | h \in [1, |I_u|]\}$ as the set of logical links in SFCR $r_u$, where $e_h^u$ indicates the $h$-th logical link between VNFs $f_h^u$ and $f_{h+1}^u$. In addition, to ensure that there will be no duplicate VNFs in one SFCR, we assume that a specific type of VNF is requested at most once in each SFCR. To avoid issues such as routing loops and the overload on a single node in the network, we assume that a VNF instance can only serve one type of VNFR. Therefore, VNFRs in one SFCR cannot be allocated to the same VNF instance.

### 3.3. VNF Forward Graph

We use a directed graph $\bar{G} = (\bar{N}, \bar{L})$ to denote VNF forward graph (VNF-FG). Intuitively, a VNF-FG is a virtual network abstracted from the physical network and consists of multiple VNF instances. The parameters $\bar{N} = \{\bar{n}_{i'} | i' \in [1, |\bar{N}|]\}$ and $\bar{L} = \{\bar{l}_{j'} | j' \in [1, |\bar{L}|]\}$ stand for the set of VNF instances and virtual links, respectively. Likewise, we use $\bar{n}_{i'} \in \bar{N}$ and $\bar{l}_{j'} \in \bar{L}$ to indicate the $i'$-th VNF instance node and $j'$-th virtual link.

The serving capability (in cycles/sec) of VNF instance $\bar{n}_{i'}$ is set to $C_{\bar{n}_{i'}}$. Moreover, considering that the resource requests of different SFCs are independent, we denote the processing density (in cycles/bit) of a VNF instance serving a SFCR $r_u$ as $w^u_{\bar{n}_{i'}}$ [24]. Therefore, the processing rate $v^u_{\bar{n}_{i'}}$ allocated by VNF instance $\bar{n}_{i'}$ to SFCR $r_u$ can be calculated as follows:

$$v^u_{\bar{n}_{i'}} = \frac{C_{\bar{n}_{i'}} \eta^u_{\bar{n}_{i'}}}{w^u_{\bar{n}_{i'}}} \quad \forall \bar{n}_{i'} \in \bar{N}, \tag{1}$$

where $\eta^u_{\bar{n}_{i'}}$ is the CPU sharing rate of the VNF instance $\bar{n}_{i'}$ serving SFCR $r_u$.

## 4. Problem Formulation

In this section, we formulate the VNF migration problem as a mathematical model. All the notations and variables are listed in Table 2.

**Table 2.** Symbols and variables.

| Symbols and Variables | Description |
| --- | --- |
| Physical network | |
| $N$ | Set of physical nodes, $n_i \in N$ is a physical node. |
| $L$ | Set of physical links, $l_j \in L$ is a physical link. |
| $C^{cpu}_{n_i}$, $C^{mem}_{n_i}$ | Capacity of CPU and memory in node $n_i$. |
| $B_{l_j}$ | Bandwidth capacity on link $l_j$. |
| $D_{l_j}$ | Propagation delay on link $l_j$. |
| SFCR related | |
| $R$ | Set of SFCRs, $r_u \in R$ is an SFCR. |
| $F_u$ | Set of VNFRs in SFCR $r_u$, $f^u_v \in F_u$. |
| $E^u$ | Set of logical links in SFCR $r_u$, $e^u_h \in E^u$. |
| $\lambda_u$ | Traffic arrival rate in SFCR $r_u$. |
| $C^{cpu}_{f^u_v}$, $C^{mem}_{f^u_v}$ | CPU and memory requirement of VNF $f^u_v$. |
| $D^{max}_u$ | The maximum tolerated delay of SFCR $r_u$. |
| VNF-FG related | |
| $\bar{N}$ | Set of VNF instance nodes, $\bar{n}_{i'} \in \bar{N}$. |
| $\bar{L}$ | Set of virtual links, $\bar{l}_{j'} \in \bar{L}$. |
| $v^u_{\bar{n}_{i'}}$ | Processing rate allocated by VNF instance $\bar{n}_{i'}$ to SFCR $r_u$ |
| Unknown variables | |
| $x^{\bar{n}_{i'}}_{f^u_v}$ | Whether VNFR $f^u_v$ is mapped on VNF $\bar{n}_{i'}$. |
| $x^{\bar{l}_{j'}}_{e^u_h}$ | Whether logical link $e^u_h$ is mapped on virtual link $\bar{l}_{j'}$. |
| $y^{n_i}_{\bar{n}_{i'}}$ | Whether VNF $\bar{n}_{i'}$ is host on node $n_i$. |
| $y^{l_j}_{\bar{l}_{i'}}$ | Whether virtual link $\bar{l}_{i'}$ is host on link $l_j$. |

First, we define the binary variable $x^{\bar{n}_{i'}}_{f^u_v}$ to indicate the mapping status of SFCR $r_u$:

$$x^{\bar{n}_{i'}}_{f^u_v} = \begin{cases} 1 & \text{VNFR } f^u_v \text{ in SFCR } r_u \text{ is mapped on VNF instance } \bar{n}_{i'}, \\ 0 & \text{otherwise.} \end{cases} \tag{2}$$

Likewise, the mapping status of logical links in SFCR $r_u$ is:

$$x^{\bar{l}_{j'}}_{e^u_h} = \begin{cases} 1 & \text{logical link } e^u_h \text{ in SFCR } r_u \text{ is mapped on virtual link } \bar{l}_{j'}, \\ 0 & \text{otherwise.} \end{cases} \tag{3}$$

Next, we define Equations (4) and (5) to denote the mapping status of a VNF-FG:

$$y_{\bar{n}_{i'}}^{n_i} = \begin{cases} 1 & \text{node } n_i \text{ holds VNF instance } \bar{n}_{i'}, \\ 0 & \text{otherwise.} \end{cases} \tag{4}$$

$$y_{\bar{l}_{j'}}^{l_j} = \begin{cases} 1 & \text{link } l_j \text{ holds virtual link } \bar{l}_{j'}, \\ 0 & \text{otherwise.} \end{cases} \tag{5}$$

The total delay of an SFCR consists of the processing delay $P_u$ on VNF instance and the transmission delay $T_u$ on links. We apply the M/M/1 queuing model to each VNF instance node. The traffic of SFCR follows the Poisson distribution with the arrival rate of $\lambda_u$ bps, while the processing time of the traffic on the VNF instance obeys the exponential distribution. Based on Little's law, the processing delay of a single VNF instance is calculated as:

$$P_u^{\bar{n}_{i'}} = \frac{x_{f_v^u}^{\bar{n}_{i'}}}{v_{\bar{n}_{i'}}^u - \lambda_u + \varepsilon} \quad \forall f_v^u \in F^u, \forall \bar{n}_{i'} \in \bar{N}, \forall r_u \in R \tag{6}$$

and the total processing delay of an SFCR is

$$P_u = \sum_{f_v^u \in F^u} \sum_{\bar{n}_{i'} \in \bar{N}} P_u^{\bar{n}_{i'}}, \quad \forall r_u \in R. \tag{7}$$

In Equation (6), $0 < \varepsilon \ll$ is a constant that prevents $P_u^{\bar{n}_{i'}}$ from being undefined, and $x_{f_v^u}^{\bar{n}_{i'}} = 0$ leads to an invalid value of $P_u^{\bar{n}_{i'}}$.

As shown in Equation (8),

$$T_u = \sum_{e_h^u \in E^u} \sum_{\bar{l}_{j'} \in \bar{L}} \sum_{l_j \in L} x_{e_h^u}^{\bar{l}_{j'}} y_{\bar{l}_{j'}}^{l_j} D_{l_j} \tag{8}$$

where $T_u$ represents the delay generated by the traffic of SFCR $r_u$ transmitted on physical links.

For SFCR $r_u$, its total end-to-end delay must meet the constraint of the maximum tolerated delay as

$$D_u = P_u + T_u \leqslant D_u^{max}. \tag{9}$$

For an SFCR $r_u$, we can only map its VNFRs and logical links once in the physical network. Then, we define the following constraints:

$$\sum_{\bar{n}_{i'} \in \bar{N}} \sum_{n_i \in N} x_{f_v^u}^{\bar{n}_{i'}} y_{\bar{n}_{i'}}^{n_i} = 1, \forall f_v^u \in F^u, \tag{10a}$$

$$\sum_{\bar{l}_{j'} \in \bar{L}} \sum_{l_j \in L} x_{e_h^u}^{\bar{l}_{j'}} y_{\bar{l}_{j'}}^{l_j} = 1, \forall e_h^u \in E^u. \tag{10b}$$

Next, we should ensure that the consumption of CPU, memory, and bandwidth cannot exceed the resource capacity on corresponding nodes and links. So, we define the following constraints:

$$\sum_{r_u \in R} \sum_{f_v^u \in F^u} \sum_{\bar{n}_{i'} \in \bar{N}} x_{f_v^u}^{\bar{n}_{i'}} y_{\bar{n}_{i'}}^{n_i} C_{f_v^u}^{cpu} \leqslant C_{n_i}^{cpu}, \forall n_i \in N \tag{11a}$$

$$\sum_{r_u \in R} \sum_{f_v^u \in F^u} \sum_{\bar{n}_{i'} \in \bar{N}} x_{f_v^u}^{\bar{n}_{i'}} y_{\bar{n}_{i'}}^{n_i} C_{f_v^u}^{mem} \leqslant C_{n_i}^{mem}, \forall n_i \in N \tag{11b}$$

$$\sum_{r_u \in R} \sum_{e_h^u \in E^u} \sum_{\bar{l}_{j'} \in \bar{L}} x_{e_h^u}^{\bar{l}_{j'}} y_{\bar{l}_{j'}}^{l_j} \lambda_u \leqslant B_{l_j} \ \forall l_j \in L. \tag{11c}$$

Considering the characteristics of the M/M/1 queue, the processing rate $v_{\bar{n}_{i'}}^u$ allocated on VNF $\bar{n}_{i'}$ can neither exceed the maximum processing rate on VNF nor be lower than the traffic arrival rate of SFCR $r_u$, Equation (12) must be satisfied

$$x_{f_v^u}^{\bar{n}_{i'}} \lambda_u \leqslant v_{\bar{n}_{i'}}^u \leqslant x_{f_v^u}^{\bar{n}_{i'}} \frac{C_{\bar{n}_{i'}}}{w_{\bar{n}_{i'}}^u}, \quad \forall r_u \in R, \forall \bar{n}_{i'} \in \bar{N}. \tag{12}$$

Then we formulate the load fluctuation in network after VNF migration. We use the resource variance to reflect the load balance status of the network [25]. That is, the smaller the resource variance, the stronger the load-balancing capability of the network. In addition, we calculate the load change as a percentage to guarantee that the optimization objectives are of the same magnitude.

As shown in Equation (13), we use notation $Q_{n_i}^*$ to indicate the resource consumption on node

$$Q_{n_i}^* = \sum_{f_v^u \in F^u} \sum_{\bar{n}_{i'} \in \bar{N}} x_{f_v^u}^{\bar{n}_{i'}} y_{\bar{n}_{i'}}^{n_i} C_{f_v^u}^*, \quad \forall n_i \in N, \tag{13}$$

where $*$ denote the resource type (CPU and memory).

Then, we define Equations (14a) and (14b) to calculate the mean and variance of node load.

$$Q_{mean}^* = \frac{\sum_{n_i \in N} \frac{Q_{n_i}^*}{C_{n_i}^*}}{|N|} \tag{14a}$$

$$Q_{var}^* = \frac{\sum_{n_i \in N} (Q_{n_i}^* - Q_{mean}^*)^2}{|N|} \tag{14b}$$

Similarly, we have the following link bandwidth consumption, mean and variance of link load

$$S_{l_j} = \sum_{e_h^u \in E^u} \sum_{\bar{l}_{j'} \in \bar{L}} x_{e_h^u}^{\bar{l}_{j'}} y_{\bar{l}_{j'}}^{l_j} \lambda_u \quad \forall l_j \in L \tag{15a}$$

$$S_{mean} = \frac{\sum_{l_j \in L} \frac{S_{l_j}}{B_{l_j}}}{|L|} \tag{15b}$$

$$S_{var} = \frac{\sum_{l_j \in L} (S_{l_j} - S_{mean})^2}{|L|} \tag{15c}$$

So, the total network load $\mathbb{L}$ is

$$\mathbb{L} = Q_{var}^{cpu} + Q_{var}^{mem} + S_{var} \tag{16}$$

We need to find the best migration solution after the migration (at time ($t$)) based on the mapping status of the VNFs before migration (at time ($t-1$)). Equations (17b) and (17a) indicate the fluctuation in network load and SFCRs delay

$$\Delta \mathbb{L}(t) = \mathbb{L}(t) - \mathbb{L}(t-1) \tag{17a}$$

$$\Delta \mathbb{D}(t) = \sum_{r_u \in R} \{D_u(t) - D_u(t-1)\} \tag{17b}$$

Our objective function is expressed as

$$Minimize \quad \omega_1 \Delta \mathbb{D}(t) + \omega_2 \Delta \mathbb{L}(t)$$
$$s.t. \; Equation \; (2) \; to \; Equation \; (17) \tag{18}$$

We aim to minimize the network load and SFC delay variation after the VNF migration is completed. Considering that there are two optimization objectives, we utilize a weighted sum approach to normalize the magnitudes of the two objectives while reflecting their relative importance.

## 5. Proposed Algorithm

Based on the problem formulation in Sections 3 and 4, we propose a Deep-Learning-based Two-Stage Algorithm (DLTSA). We first utilize the Hybrid Genetic Evolution Algorithm to derive the training data based on available resources for our problem. Then, we apply the running algorithm, which can effectively derive the VNF migration schemes.

### 5.1. Genetic Evolution on VNF Migration

The Genetic Evolution (GE) Algorithm is an evolutionary algorithm that simulates bio-genetics [26]. They adapt to the environment by employing the mechanisms of population evolution (inducing operators such as mutation and crossover between individuals) [27]. When a VNFI is migrated, it may influence multiple SFCs. To simplify this problem, we regard the potential migration states of all VNF instances in the network as an entire solution. The GE algorithm mainly consists of four essential parts:

#### 5.1.1. Initialization

First, we generate an initial population $F(0) = \{x_{1,0}, x_{2,0}, ..., x_{n,0}\}$ with $n$ individuals, where each $x_{i,0}$ represents a possible migration plan for this network. Then, if there exist $m$ VNFs to be migrated, a complete solution $x_{i,0}$ is a vector composed of $m$ elements. For example, $x_{i,0} = \{n_2, n_4, n_5\}$ means that the first, second and third VNFs are migrated to the physical nodes $n_2$, $n_4$ and $n_5$.

#### 5.1.2. Fitness Calculation

Based on the current population $F(t)$, we calculate the fitness function for each $x_{i,t}$. The algorithm employs the fitness function to select the next generation of individuals and then seeks the optimal solution to the problem. An individual represents a possible migration scheme for the VNF migration problem. The fitness function of each individual in current population to fit in with the survival of the fittest. Therefore, the definition of the fitness function is critical, and it is related to the algorithm's convergence speed and the solution's quality. We exploit the idea of the penalty function to transform the objective function. Our fitness function is the sum of two parts. The former part is the normalized value of objective, and the recent past is the penalty function. The specific equation is as follows:

$$f(x_{i,t}) = \omega_1 \Delta \mathbb{D}(t) + \omega_2 \Delta \mathbb{L}(t) + \delta * S(x_{i,t}) \tag{19a}$$

$$S(x_{i,t}) = \sum_{n_i \in x_{i,t}} max\{0, \sum_{\bar{n}_i \longrightarrow n_i} y_{\bar{n}_{i'}}^{n_i} Q_{\bar{n}_{i'}}^{*} - \rho_{n_i}^{*}\} \tag{19b}$$

$$Q_{\bar{n}_{i'}}^{*} = \sum_{r_u \in R} \sum_{f_v^u \in F^u} x_{f_v^u}^{\bar{n}_{i'}} C_{f_v^u}^{*} \tag{19c}$$

where $\delta$ is the penalty factor, and $S(x_{i,t})$ represents the penalty strength for choosing candidate solution $x_{i,t}$. The parameter $\rho_{n_i}^{*}$ in Equation (19b) is the remaining resource on node $n_i$. Equation (19c) is the total resources required by VNF $\bar{n}_{i'}$. For Equation (19b), if Equation (19c) is greater than $\rho_{n_i}^{*}$, the value of $S(x_{i,t})$ is greater than zero, thus triggering the penalty. The penalty strength depends on whether the current candidate migration

plan satisfies the node resource constraints. If the node resource limit is satisfied, there is no penalty. When the node limit is violated, the more the current scheme exceeds the constraints, the greater the penalty strength. In addition, we assign a large value ($\delta = 15$) to the penalty factor to filter out feasible solutions more quickly.

### 5.1.3. Individual Selection

Based on the fitness of individuals in population $F(t)$, we use a roulette to determine which individuals will be inherited by $F(t + 1)$. The fitness value is proportional to the probability of individual selection, and the probability $P(i)$ of selecting the $i$-th individual is as follows:

$$P(x_{i,t}) = \frac{f(x_{i,t})}{\sum_{x_{i,t} \in F(t)} f(x_{i,t})} \quad i = (1, 2, ..., n) \tag{20}$$

### 5.1.4. Crossover and Mutation

Crossover and mutation are operations that generate new individuals to maintain the diversity of the population. They cooperate to complete the global search for the solution space. We use the Partial-Mapped Crossover method to recombine the two individuals. The probabilities of these two actions are denoted as Equations (21a) and (21b)

$$P_c = \begin{cases} \frac{k_1(f_{max} - f)}{f_{max} - f_{avg}} & f \geq f_{avg} \\ k_2 & f < f_{avg} \end{cases} \tag{21a}$$

$$P_m = \begin{cases} \frac{k_3(f_{max} - f')}{f_{max} - f_{avg}} & f' \geq f_{avg} \\ k_4 & f' < f_{avg} \end{cases} \tag{21b}$$

$f_{avg}$ and $f_{max}$ represent the population's average fitness and maximum fitness, respectively. The parameter $f_{avg}$ ($f_{max}$) denotes the higher fitness of the two individuals to be crossed (mutated).

The procedure of HGE algorithm is given in Algorithm 1. First, based on the size of resource consumption, we sort the VNF instances that need to be migrated in descending order and generate individual gene rankings (lines 1–2 in Algorithm 1). Some invalid individuals may be generated during population initialization and iteration. In these invalid solutions, the total resource consumption of the VNFs migrated to the target node exceeds the available resources on the node. To improve the algorithm's efficiency and resolution quality, we add the heuristic search algorithm BFD (Best Fit Decreasing) in Algorithm 1 as a pre-stage. The BFD algorithm is called at line 3 of Algorithm 1, and the specific content is introduced in the following subsection. Algorithms 1 and 2 together constitute the DLTSA algorithm.

At line 8 in Algorithm 1, we execute the crossover and mutation operations on the population, which may probabilistically generate some invalid individuals. This may result in the violation of delay or resource constraints. To improve population effectiveness, we need to fix all invalid individuals. In Algorithm 1, line 9 involves a partial update where the algorithm filters out individuals whose fitness is affected by the penalty factor. These individuals are decoded to create a set of solutions. If an invalid solution is detected, the algorithm assigns the incorrect VNF instances in descending order to the nodes in the adjacent rack where the VNF instance is located. This operation is performed before migration to meet the SFC delay constraints and ensure sufficient resources are available. Therefore, we correct the invalid individual and inherit them to a new population $F(t + 1)$, guaranteeing the validity of the population (lines 9–10 in Algorithm 1).

---

**Algorithm 1** Hybrid genetic evolution algorithm.

---

**Input:** Physical network $G = (N, L)$, VNF-FG $\bar{G} = (\bar{N}, \bar{L})$, Set of SFCRs $R$, $k_1, k_2, k_3, k_4$.
**Output:** $X_{bext,t_{max}} = \{x_{best,1}, x_{best,2}, ..., x_{best,M}\}t_{max}$.

 1: $t \leftarrow 0$.
 2: Obtain set of VNF instances $M_{mgr}$ that waits for migrating
 3: Initialize population $F(0)$ with $n$ individuals using Algorithm 2.
 4: **while** $t \leqslant t_{max}$ **do**
 5:    Calculate the fitness value $f(x)$ of each individual in the population $F(t)$ exploiting Equation (19).
 6:    **repeat**
 7:       Select two individuals based on the probability calculated by Equation (20).
 8:       Execute the crossover and mutation operations using Equation (21).
 9:       Detect and correct the validity of two individuals.
10:       Inherit two individuals to a new population $F(t+1)$.
11:    **until** $|F(t+1) = n|$
12:    $t \leftarrow t + 1$
13: **end while**
14: Find the best individual $x_{best,t_{max}}$.
15: **return** $x_{best,t_{max}} = \{x_{best,1}, x_{best,2}, ..., x_{best,M}\}$.

---

**Algorithm 2** Best fit decreasing algorithm.

---

**Input:** Physical network $G = (N, L)$, VNF-FG $\bar{G} = (\bar{N}, \bar{L})$, Set of VNF instances $M_{mgr}$ to be migrated.
**Output:** Initial population $F(0)$.

 1: Sort $M_{mgr}$ in descending order based on resource size derived by Equation (19c).
 2: $i \leftarrow 0$.
 3: **while** $i < n$ **do**
 4:    **while** VNF instance $m \in M_{mgr}$ **do**
 5:       Choose the set of migration candidate nodes that can hold VNF instance $m$.
 6:       Add the node with max available resource to $x_{i,0}$.
 7:       Update the available resources of the node.
 8:    **end while**
 9:    Add the individual to population $F(0)$ and reset $\bar{N}$.
10:    $i \leftarrow i + 1$.
11: **end while**
12: **return** $F(0)$

---

*5.2. Pre-Stage in Hybrid Genetic Evolution Algorithm*

If the fitness value of an individual is unreasonable [28], its corresponding solution violates the node's resource constraints. Therefore, Algorithm 2 is designed for reconstructing individual solutions. Algorithm 2 demonstrates the procedure of generating the initial population. The parameter $M_{mgr}$ is the group of VNF instances waiting for migration. We first arrange the VNF instances that have not been successfully migrated in descending order (line 1 in Algorithm 2). Then, we select nodes with sufficient resources from the adjacent nodes of these VNFs as their migration nodes to satisfy the resource and delay constraints (line 5 in Algorithm 2). Finally, we migrate the VNF to the target node with the most available resources to satisfy the VNF resource constraints. Meanwhile, to prevent resource overload, we update the remaining resources of the node (lines 6–7 in Algorithm 2). By recoding the invalid genes of individuals, we can ensure the efficacy of breeding new individuals in the iteration procedure.

*5.3. Running Algorithm of DLTSA*

During the process's second phase, we obtain the optimal migration node group from the training data gathered in the first phase. Once we have this group, we select the VNF instances that will be migrated.

The running process of DLTSA is presented in Algorithm 3 which includes initializing related parameters and obtaining the optimal VNF instance group in lines 1–3. In lines 4–9, we compute the optimal target node for migrating VNFs. From there, we obtain the complete migration scheme $\Psi$ in lines 10–17 if $\Psi$ satisfies all the constraints in Equations (9)–(12). It will be an output to migrate the corresponding VNFs. However, if $\Psi$ does not meet these constraints, we iterate and check other candidate nodes to recompute $\Psi$ in line 14 of Algorithm 3. The running process of DLTSA stops once there are no feasible nodes or the iteration times reach a stopping threshold.

---

**Algorithm 3** Running algorithm of DLTSA.

---

**Input:** Training data $\{x_{best,1}, x_{best,2}, ..., x_{best,M}\}$, Physical network $G = (N, L)$, VNF-FG $\bar{G} = (\bar{N}, \bar{L})$, Set of SFCRs $R$.
**Output:** Complete migration solution $\Psi$.
 1: Initialize maximum iteration times $\alpha$, current iteration times $\beta$, and migration nodes $\Psi \leftarrow \varnothing$.
 2: Generate the input parameter $x_r$ based on $C_{f_v^u}^{cpu}$, $C_{f_v^u}^{mem}$, $\lambda_u$, $D_u^{max}$.
 3: Obtain candidate nodes $\{x_{best,1}, x_{best,2}, ..., x_{best,M}\}$ of SFCR $r_u$
 4: **for all** VNF instances in $m \in M_{mgr}$ **do**
 5:     Choose the set of migration candidate nodes that can hold VNF instance $m$.
 6:     Add the node with max available resource to $x_{i,0}$.
 7:     Update the available resources of the node.
 8:     $\Psi \leftarrow$ Obtain the optimal migration node;
 9: **end for**
10: **repeat**
11:     **if** $\Psi$ satisifies all the constraints **then**
12:         **return** $\Psi$;
13:     **else**
14:         $\Psi \leftarrow$ Iterate and recompute $\Psi$;
15:         $\beta \leftarrow \beta + 1$
16:     **end if**
17: **until** $\beta > \alpha$
18: **return** Rejected;

---

## 6. Performance Evaluation

*6.1. Simulation Setup*

We perform the evaluation on a computer with Intel(R) Core(TM) i7-10870H CPU 2.20 GHz and 32GB RAM and implement all the algorithms with MATLAB 2016a.

We use a 3-layer fat-tree of k-ary Clos topology [29], and consider two network sizes ($k = 4$ and $k = 8$). These two networks contain 16 and 128 physical nodes, respectively. $\omega_1$ and $\omega_2$ are two weighted parameters, which can be adjusted to achieve any desired trade-off. We consider that load balance and SFC delay have equal importance, so we set $\omega_1 = \omega_2$. Referring to [30,31], the setting of parameters is shown in Table 3. For the training data, first, we generate a set of SFCRs with the above parameter settings and obtain the optimal solution for each SFCR one by one by running the Hybrid Genetic Evolution Algorithm until the network's available resources are exhausted. Then, repeat the previous steps until we have collected enough training data. The network simulator generated a training dataset consisting of approximately $2 \times 10^7$ items to train the deep belief network. Additionally, each deep belief network has three hidden layers with 40 hidden neurons each.

**Table 3.** Simulation parameter settings.

| Parameters | Value | Parameters | Value |
|---|---|---|---|
| $C_{n_i}^{cpu}$ | [10, 30] cores | $C_{n_i}^{mem}$ | [32, 64] GB |
| $B_{l_j}$ | [500, 1000] Mbps | $D_{l_j}$ | [2, 5] ms |
| $D_u^{max}$ | [10, 50] ms | $\lambda_u$ | [1000, 4000] packets/s |
| SFC length | [3, 5] | $k_1, k_2$ | 0.4, 0.75 |
| $k_3, k_4$ | 0.091, 0.063 | Max generations | 500 |

*6.2. Contrastive Benchmarks*

With respect to the benchmarks, we introduce the following approaches for comparison.

- To solve small-scale problems, we use Gurobi [32], a mathematical programming solver with precise algorithms like branch and bound. Gurobi can solve complex mathematical models efficiently by traversing the solution space. However, for larger networks, Gurobi's execution time increases dramatically, and it may crash before completing the task. Therefore, we only use Gurobi as a small and medium-scale network baseline.
- To compare the convergence of the DLTSA in the evolution process, we introduce the unimproved Original Genetic Evolution algorithm (OGE) as the baseline. It is important to note that the OGE algorithm generates initial individuals randomly and does not use the best-fit decreasing heuristic for optimization.
- The Backtracking-based Greedy algorithm (BG) [33] determines migration decisions based on the memory size of the migration service. It employs two variables, allocation ratio, and backtracking rate, to manage the backtracking process and minimize the allocation of nodes with limited resources.

*6.3. Simulation Results*

In the working phase of SFCs, the flow of services constantly changes, which may lead to an overload of network nodes and links, and induces end-to-end delay conflicts of services. Load balancing can make the network more tolerant to future traffic changes, and ensuring the end-to-end delay of SFC after VNF migration can effectively improve QoS.

6.3.1. Fitness Value

Regarding the fitness value, we can see from Figure 2 and 3 that the DLTSA algorithm outperforms the OGE algorithm. The reason is that numerous invalid individuals flood the evolution of the OGE algorithm, affecting the population's effective iteration. In addition, as the network size increases, the convergence speed of the algorithms decreases. When the network size is $k = 4$, the DLTSA and OGE algorithms are steady after about 200 and 250 iterations, respectively. Moreover, in a network with $k = 8$, these two algorithms need about 300 and 400 iterations to achieve stability. This is because, with the increase in network scale, the solution space increases, and the evolutionary algorithm needs to go through more iterations to find the optimal solution of the solution space.

Our approach always stays at 0 in terms of individuals' invalid ratio. However, the OGE algorithm achieves 39% and 43% in two network topologies, resulting in many invalid solutions. After removing invalid individuals, the solution quality of the GE algorithm is inferior to that of the DLTSA algorithm. Therefore, the solution quality of DLTSA is better than that of the original genetic evolution approach.

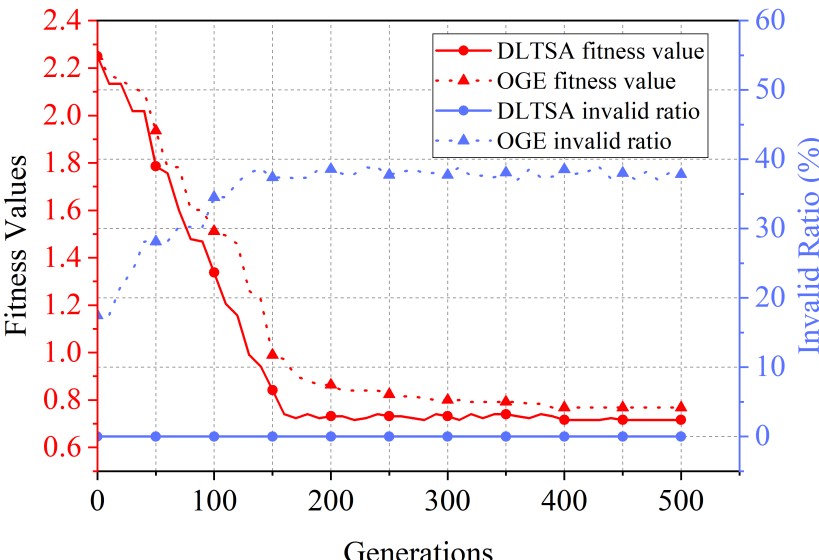

**Figure 2.** Fitness value in a 4-ary fat-tree.

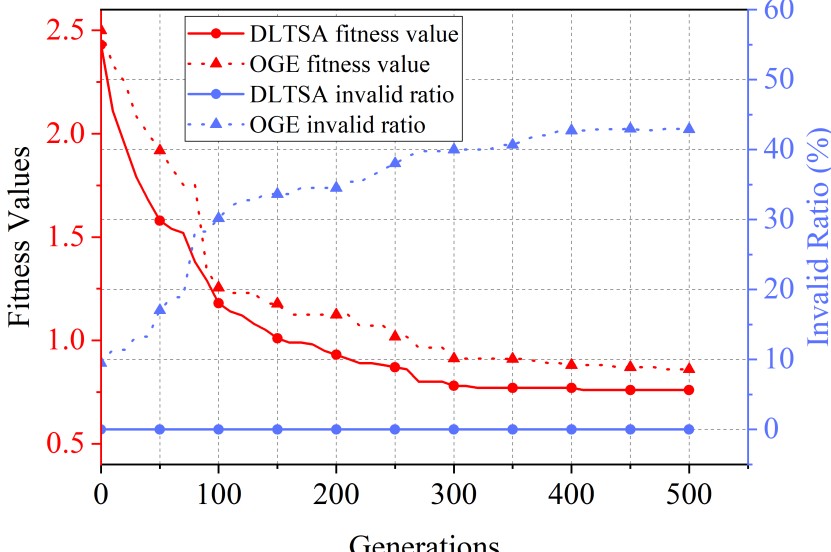

**Figure 3.** Fitness Value in a 8-ary fat-tree.

### 6.3.2. Network Load Balance

In the initial network mapping, the optimization goal is to minimize the accumulated SFC delay, which results in VNFs being placed in the same or adjacent pods and poor load balance in the network. The difference in the initial load of the three networks depends on the number of SFCs mapped. As shown in Figure 4, the DLTSA algorithm performs the best in VNF migration compared to other algorithms, including OGE and BG, and is close to the optimal solution Gurobi. This is because the essence of the BFD-integrated DLTSA algorithm is a global search. After a complete evolution process, the solution obtained is better. Although OGE and BG algorithms perform poorly, they are still better than the initial network state, proving the effectiveness of the VNF migration mechanism. By considering the sharing of VNF, DLTSA can make more efficient use of network resources.

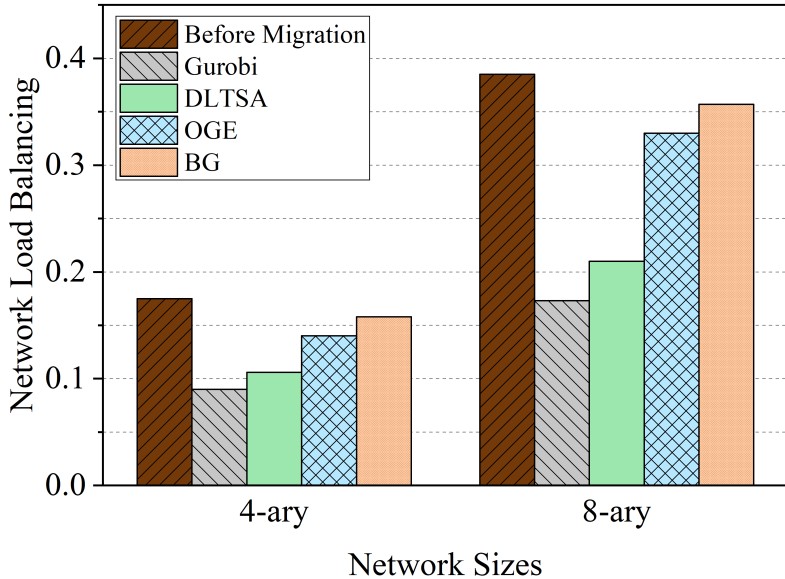

**Figure 4.** Comparison of network load balance.

6.3.3. Average SFC Delay

We can see from Figure 5 that the DLTSA algorithm has the most negligible impact on the post-migration SFC delay. Compared to the benchmarks, the average SFC delay is reduced by about 15% to 25% by DLTSA in different network sizes. This is because the DLTSA algorithm considers the sharing of VNF instances. During the iteration, the population is polluted due to the large number of invalid solutions produced by the OGE algorithm. This also leads to poor performance of OGE. In addition, the OGE algorithm does not consider the significance of simultaneous migration and does not have a rational prioritization approach. Fewer migration options are available during the later VNF migration phase, which impacts multiple SFCs. The BG algorithm also overlooks the sharing of VNFs, as a successful VNF migration for one SFC may cause significant delays for other affected SFCs. Based on the above analysis, DLTSA can obtain a better SFC delay after VNF migration than other algorithms. Based on the above analysis, DLTSA can obtain a better SFC delay after VNF migration than other algorithms.

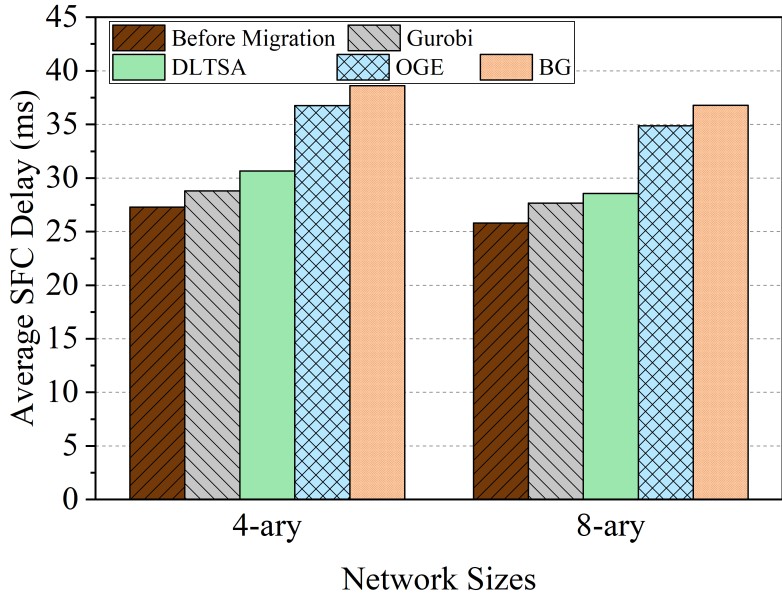

**Figure 5.** Comparison of average SFC delay.

### 6.3.4. Execution Time

According to the data presented in Figure 6, the execution time of four algorithms was analyzed by 1000 SFCRs. The results reveal that the optimal algorithm (Gurobi) consumes the longest time of approximately 3200 ms. On the other hand, OGE and BG algorithms consume around 1600 ms and 2400 ms, respectively. The reason is that OGE and BG approaches inspect constraints after obtaining VNF migration solutions and re-enter the iteration process or check other solutions if the solution violates any constraint. Interestingly, DLTSA performed exceptionally well, eight times faster than OGE and 16 times faster than the optimal algorithm. Because the DLTSA approach considers all constraints during the iteration process and employs a penalty function to effectively reduce the solution space, resulting in an improved algorithm efficiency.

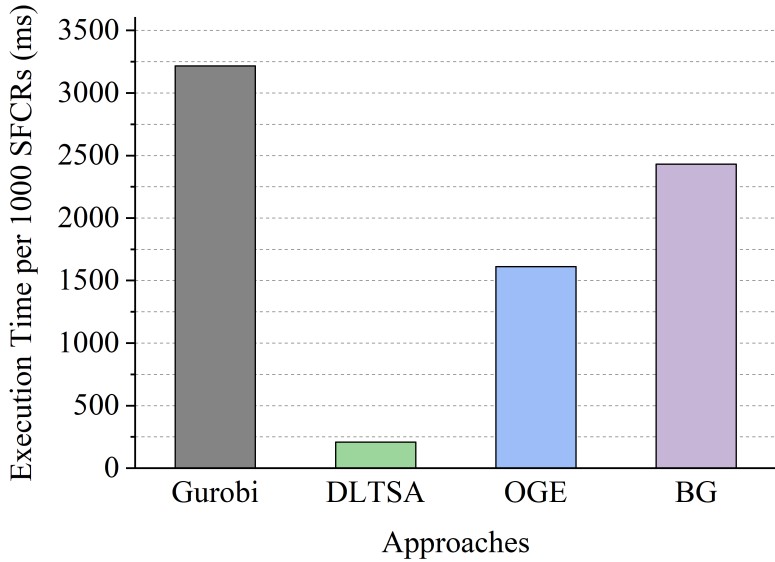

**Figure 6.** Comparison of time efficiency.

## 7. Conclusions

This paper focuses on the challenge of VNF migration and SFC reconfiguration in 6G-MEC networks. We have formulated this problem as a mathematical model that considers the shared use of VNF instances by multiple SFCs. Although this adds complexity to our problem, it aligns more with real-world requirements. We propose a Deep-Learning-based Two-Stage Algorithm (DLTSA) to address it. We also devise a running algorithm to reduce the computational burden of DLTSA. In DLTSA, we have emphasized the importance of sharing VNFI and migration concurrency. We aim to minimize the end-to-end delay for all services and achieve network load balancing after migration. Our simulation results show that these algorithms, considering VNFI sharing, can guarantee network load balancing while reducing service delay as much as possible.

However, there are limitations to our work. We have only evaluated our solutions in a simulation environment, and we plan to implement the migration model and algorithms in a real NFV platform for future work. Furthermore, we recognize that additional requirements and constraints, such as migration cost and energy consumption, need to be considered.

**Author Contributions:** Conceptualization, Y.Y. and X.T.; methodology, Y.Y. and X.T.; software, X.Z.; validation, Y.Y., W.Y. and X.Z.; formal analysis, Y.Y.; investigation, Z.Z.; resources, Y.Y.; writing—original draft preparation, Y.Y.; writing—review and editing, Y.Y. and Z.Z.; All authors have read and agreed to the published version of the manuscript.

**Funding:** This work is supported by the 2023–2025 Young Elite Scientists Sponsorship Program of Beijing Association for Science and Technology. No. BYESS2023283.

**Data Availability Statement:** Not applicable.

**Conflicts of Interest:** The authors declare no conflict of interest.

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
