# Peer review of "Virtual Network Function Migration Considering Load Balance and SFC Delay in 6G Mobile Edge Computing Networks"

_electronics, doi:10.3390/electronics12122753_

Round 1
Reviewer 1 Report
This paper addresses VNF migration in 6G MEC networks. The authors propose an optimization model to balance network load and end-to-end delay. They introduce a two-stage algorithm called DLTSA that leverages deep learning techniques, supported by simulation results.
Overall, this paper is well-organized and relatively easy to read, with a clear problem statement. However, there are areas that need improvement:
-
Certain terms, such as VNF migration, fitness function, and individuals in the proposed algorithm, need better definitions. Additionally, SFCR is used before being defined. Furthermore, the authors should clarify their choice of Mobile Edge Computing (MEC) in 6G mobile networks.
-
Figure 1 does not effectively explain the proposed solution. It would be beneficial for the authors to replace it with a more informative figure that illustrates how DLTSA works.
-
The simulation does not appear to use real traffic. The authors should explain if this choice affects the results and how they mitigate this issue. Certain goals, such as minimizing network load and SFC delay variation after VNF migration, require a clearer explanation of their importance in the given scenario.
-
Annotating the conditions on page 5 would enhance the clarity of the equations used in the paper.
-
More citations should be provided for algorithms like the Genetic Evolution (GE) algorithm, which may be well-known in the AI field but not as familiar in computer networks, particularly to readers new to this field.
-
The algorithm does not seem to consider the priority of VNFs in the calculations, which is a limitation.
-
On page 11, Algorithm 5 is mentioned but not included. This may be a typo.
-
In the evaluation, the authors should clarify the reasoning behind their choice of algorithms and define parameters properly, such as "w1=w2." Figures should be distinguishable in black-and-white copies, and their titles should be self-explained. Deep insights should be provided when making statements, such as explaining why DLTSA performed exceptionally well.
No major issues
Author Response
Based on the suggestions, we have carefully revised the paper and provided point-to-point response in this letter. Please see the attachment.

Reviewer 2 Report
This paper presents a well-organized structure related to NFV-enabled networks and detailed backgrounds of VNFFG and SFC. The variables and parameters in VNF placement are described in detail, which is helpful for the readers. In the proposed method, the authors clearly point out the formulation of VNF migration and the objective function of minimizing the E2E delay of all instantiated SFC. The problems are handled by an intelligent deep learning-based two-stage algorithm (hybrid genetic evolution and running algorithm). The performance evaluation is well-presented; however, there are several concerns to address as follows.
1. In algorithm 2 and 3, within the VNF instance loops, the operation always add the node with max available resource to x(i,0). Is it reliable to add all the nodes to a stacked index-i without iteration/rotating? Is x(i,0) a relevant output from x(best,tmax)?
2. Section 5 should emphasize the optimization process to achieve the network load and SFC delay minimization, including a specification of optimizing the penalty strength and factor.
3. The performance evaluation should present the hyper-parameter configuration from deep learning perspectives.
4. The following papers are recommended to cite (optionally), it is related to the NFV-enabled networks and Deep Learning-related simulation:
- Tam, P.; Math, S.; Kim, S. Priority-Aware Resource Management for Adaptive Service Function Chaining in Real-Time Intelligent IoT Services. Electronics 2022, 11, 2976.
- Pei, J.; Hong, P.; Pan, M.; Liu, J.; Zhou, J. Optimal VNF placement via deep reinforcement learning in SDN/NFV-enabled networks. IEEE J. Sel. Areas Commun. 2019, 38, 263–278.
- Tam, P.; Math, S.; Kim, S. Optimized Multi-Service Tasks Offloading for Federated Learning in Edge Virtualization. IEEE Trans. Netw. Sci. Eng. 2022, 1–17.
Author Response
Based on the suggestions, we have carefully revised the paper and provided point-to-point response in this letter. Please refer to the following detailed comments and responses. Please see the attachment.

Reviewer 3 Report
In general, scientific of this paper is acceptable and the topic itself is still highly relevant. However, the quality and structure of the submitted article should be significantly improved, as many parts of the paper (apart from method description and experiments) seem incomplete, not informative enough or too short. The following modifications should be taken into account:- Too few references for such journal publication. Please take into account more related works, as there are a lot of them available.
- Try to better distinguish what should be part of related works and what goes to introduction. Currently, these two sections somehow are not well distinguished. Try to put more emphasis on the challenges and issues tackled, together with practical perspective in the introudction.
- Related works section can include a tabular summary/overview of the existing relevant approaches/solutions, emphasizing the case of studies, applied methods and results
- Evaluation part should be more detailed when it comes to discussion of relevant aspects (execution time, achieved benefits) and comparison to relevant approaches can be taken into account as well
- Put figures as close as possible after description of them, in the same (sub)section
- The length of conclusion is just unacceptable. More observations could be taken into account, comparison to similar approaches (summarized) and future works as well.
- In both introduction and conclusion, real-world aspects and implications related to solving this problem should be taken into account, putting the reader's attention to more practical perspective
No major issues detected, quite simple style, easy to follow, while proofreading is recommended. Please try to avoid sentences such as "But little literature has paid attention to this issue." and put them together into more specific context with additional information.
Author Response

(The authors gave the same response as above.)

Round 2
Reviewer 3 Report
Thank you for incorporating the requested changes. The quality of the paper is now fine for publication.
No major issues detected, quite simple style, easy to follow, while proofreading is recommended. Please try to avoid sentences such as "But little literature has paid attention to this issue." and put them together into more specific context with additional information.